# Hashimoto’s Thyroiditis and Graves’ Disease in Genetic Syndromes in Pediatric Age

**DOI:** 10.3390/genes12020222

**Published:** 2021-02-04

**Authors:** Celeste Casto, Giorgia Pepe, Alessandra Li Pomi, Domenico Corica, Tommaso Aversa, Malgorzata Wasniewska

**Affiliations:** Department of Human Pathology of Adulthood and Childhood, Unit of Pediatrics, University of Messina, 98124 Messina, Italy; celestecasto@libero.it (C.C.); giorgiapepe23@gmail.com (G.P.); alessandra.lipomi92@gmail.com (A.L.P.); dcorica@unime.it (D.C.); tommaso.aversa@unime.it (T.A.)

**Keywords:** Hashimoto’s thyroiditis, Graves’ disease, genetic syndromes, Turner syndrome, Down syndrome, Klinefelter syndrome, RASopathies, 22q11.2 deletion syndrome, Prader–Willi syndrome, Williams syndrome

## Abstract

Autoimmune thyroid diseases (AITDs), including Hashimoto’s thyroiditis (HT) and Graves’ disease (GD), are the most common cause of acquired thyroid disorder during childhood and adolescence. Our purpose was to assess the main features of AITDs when they occur in association with genetic syndromes. We conducted a systematic review of the literature, covering the last 20 years, through MEDLINE via PubMed and EMBASE databases, in order to identify studies focused on the relation between AITDs and genetic syndromes in children and adolescents. From the 1654 references initially identified, 90 articles were selected for our final evaluation. Turner syndrome, Down syndrome, Klinefelter syndrome, neurofibromatosis type 1, Noonan syndrome, 22q11.2 deletion syndrome, Prader–Willi syndrome, Williams syndrome and 18q deletion syndrome were evaluated. Our analysis confirmed that AITDs show peculiar phenotypic patterns when they occur in association with some genetic disorders, especially chromosomopathies. To improve clinical practice and healthcare in children and adolescents with genetic syndromes, an accurate screening and monitoring of thyroid function and autoimmunity should be performed. Furthermore, maintaining adequate thyroid hormone levels is important to avoid aggravating growth and cognitive deficits that are not infrequently present in the syndromes analyzed.

## 1. Introduction

Autoimmune thyroid diseases (AITDs) are the most common autoimmune diseases, affecting approximately 5% of the general population, with a female gender predominance [1]. Although their prevalence reaches a peak in adulthood, AITDs are the most common cause of acquired thyroid disorder during childhood and adolescence. AITDs includes Hashimoto’s thyroiditis (HT) and Graves’ disease (GD), which are both characterized by infiltration of the thyroid by T and B lymphocytes that react against thyroid antigens and the production of thyroid autoantibodies. The main thyroid autoantibodies are directed against thyroid peroxidase (anti-TPO), thyroglobulin (anti-TG) and thyroid-stimulating hormone receptors (TRABs) [2]. The autoimmune injury of the gland is responsible for clinical and biochemical alterations (hyperthyroidism in GD and hypothyroidism in HT). In some genetic syndromes, AITDs show some peculiarities in terms of epidemiology, pathophysiology and clinical course, especially when they arise during childhood. The aim of this systematic review is to summarize the main features of AITDs when they occur in association with some genetic syndromes. 

### 1.1. Genetic Susceptibility in Autoimmune Thyroid Diseases

The etiology of AITDs is complex and multifactorial, since environmental and genetic factors act in synergy to promote thyroid autoimmunity. Genetic factors are responsible for 80% of the risk of developing AITDs, while the remaining 20% is due to environmental factors (such as iodine, drugs, infection, stress, radiation) [3]. Epidemiological evidence of genetic susceptibility to AITDs is provided by familial clustering of the disease (20–30% of cases show a positive family history of AITDs) [4].

Several genes have been identified for AITDs: some genes are specific for GD or HT, others are common to both diseases or non-thyroidal autoimmune disorders [5]. The majority of the susceptibility genes are involved in immune-regulatory processes such as central tolerance, peripheral tolerance, antigen presentation and lymphocyte activation. Alterations in the activity of these genes due to polymorphisms can lead to autoimmunity development [6].

The major genes related to AITDs include thyroid-specific genes and immune-regulatory genes:

*Thyroid-specific genes:* Considering the pathogenesis of GD, the thyroid-stimulating hormone receptor (TSHR) gene, located on chromosome 14q31, was thought to be a possible candidate gene of disease. Genome-wide association studies (GWASs) confirm this theory [7,8]. Consecutive comprehensive sequence analyses of the TSHR gene localized the causative variant within intron 1 and five single nucleotide polymorphisms (SNPs) associated with GD were identified. These SNPs may reduce the expression of the TSHR gene in the thymus and impair central tolerance [9]. 

Thyroglobulin (TG) accounts for 80% of thyroid protein content and represents an important target in AITDs. Linkage studies demonstrated an association between AITDs and a locus on chromosome 8q, containing the TG gene. Sequencing of the TG gene has shown several SNPs, strongly associated with AITDs, that caused amino acid changes in TG. Some studies have found a statistical interaction between TG gene variants and the HLA-DR variant, containing an arginine at position β74. HLA-DR is a class II HLA gene that plays a critical role in antigen presentation. Considering the statistical interaction between the two genes, some authors have hypothesized that the HLA-DR variant may present pathogenic TG peptides and trigger AITDs. However, the link between TG SNPs and the production of pathogenic TG peptides has not been established to date [6,10,11,12].

*Immunoregulatory genes:* FOXP3 is a gene localized on the X chromosome and is involved in the control of regulatory T cells (Tregs). Mutations in FOXP3 cause a rare disease characterized by immune dysregulation, polyendocrinopathy and enteropathy (IPEX syndrome) [13]. Several FOXP3 polymorphisms are associated with both HT and GD [14,15]. 

Other genes that seem to be associated with AITDs, in particular with GD, are CD25 and CD40 genes, located on chromosome 10p15 and 22q11, respectively [16]. CD40, a tumor necrosis factor receptor, plays a relevant role in adaptive immunity, being involved in B cell proliferation, immunoglobulin class switching and crosstalk between antigen-presenting cells (APCs) and T cells. The causative polymorphism that could predispose to GD (rs1883832) leads to an upregulation of CD40, which lowers the threshold for B cell activation [6,17]. 

CTLA-4 and PTPN22, located on chromosome 2q33 and 1p13, respectively, are negative regulators of T cell activation and polymorphisms of these genes are associated with AITDs [18,19,20,21].

HLA-DR, in particular the DRb1-Arg74 HLA variant, is the main susceptibility gene for GD. Some reports suggest that HLA-DR can also predispose to HT, but the results are less conclusive [22,23].

### 1.2. Autoimmune Thyroid Diseases in Pediatric General Population

AITDs are the most common thyroid disease in the pediatric population, they usually occur during puberty, are more prevalent in females and show familiar clustering [24,25]. Specifically, HT and GD prevalence rates are 1.2% and 1%, respectively [26,27]. Diagnosis is established by detecting serum thyroid autoantibodies and structural alteration on the thyroid ultrasound scan [2].

The main features of GD presentation are related to hyperthyroidism, whereas HT at diagnosis shows variable degrees of thyroid function impairment. Euthyroidism is by far the most common initial pattern (about 52% of patients), followed by overt hypothyroidism (22.2%), subclinical hypothyroidism (SH, 19.2%) and hyperthyroidism (about 6.5%) [28]. Conversion of HT into GD has been observed in at least 3–4% of children and adolescents [29]. The natural history of AITDs in the pediatric population is not fully known. In HT children presenting SH at diagnosis, there is a high risk of deterioration over time of thyroid function, if compared to those presenting with euthyroidism. Nevertheless, a spontaneous normalization of thyroid-stimulating hormone (TSH) serum values could occur in around 40% of SH patients [30]. In HT patients who display an initial picture of hyperthyroidism, either overt or subclinical, a spontaneous normalization of thyroid function is generally observed [31].

## 2. Materials and Methods

A systematic research was carried out, covering the last 20 years (from January 2000 to October 2020), according to the PRISMA statement [32], through MEDLINE via PubMed and EMBASE databases to identify studies reporting a correlation between AITDs and genetic syndromes in children and adolescents. The research was based on the combination of two or more of the following keywords to generate a wide search: (“Thyroid OR Autoimmune Thyroid Diseases OR Autoimmune Thyroiditis OR Hashimoto’s thyroiditis OR Graves’ disease”) AND (“Genetic Syndrome OR Down Syndrome OR Turner syndrome OR Klinefelter Syndrome OR Prader-Willi Syndrome OR 22q11.2 Deletion Syndrome OR Neurofibromatosis OR Noonan Syndrome OR Williams Syndrome OR RASopathies OR Chromosomopathies OR Deletion Syndromes OR Imprinting Defect” AND “Pediatric OR Children OR Adolescent”). These terms were combined in various ways to generate a wide search. 

The inclusion criteria were: articles written in English, belonging to the categories of clinical study, clinical trial, clinical trial protocol, multicenter study, randomized controlled trial and observational study, which report an association with AITDs (assessed by biochemical evaluation) and confirmed genetic syndromes in a pediatric population (age < 18 years). Due to the rarity of GD and some genetic syndromes, case reports were also included.

The exclusion criteria were: articles belonging to the categories of review, systematic review and meta-analysis or which included only a non-pediatric population (age ≥ 18 years) and/or did not evaluate AITDs or confirmed genetic syndromes.

From our initial research, all the genetic syndromes most frequently associated with AITDs were subsequently reviewed individually, hence, we focused on: Turner syndrome, Klinefelter syndrome, Down syndrome, 18q deletion syndrome, Prader–Willi syndrome, 22q11.2 deletion syndrome, neurofibromatosis type 1, Noonan syndrome and Williams syndrome. 

Titles and abstracts of all retrieved articles were screened by one author (C.C.) to identify articles for full-text review. The eligibility of all full-text articles was assessed by all authors.

## 3. Results

The initial search resulted in 1654 references. Articles that did not report an association between AITDs and genetic syndromes were excluded on the basis of title/abstract (1147 articles). The remaining records were screened for inclusion criteria. After an additional, comprehensive analysis and the exclusion of duplicates, 114 potentially eligible studies were retrieved for full-text screening. 

For our final evaluation, we selected 90 articles [33,34,35,36,37,38,39,40,41,42,43,44,45,46,47,48,49,50,51,52,53,54,55,56,57,58,59,60,61,62,63,64,65,66,67,68,69,70,71,72,73,74,75,76,77,78,79,80,81,82,83,84,85,86,87,88,89,90,91,92,93,94,95,96,97,98,99,100,101,102,103,104,105,106,107,108,109,110,111,112,113,114,115,116,117,118,119,120,121,122], which were included in the review (Figure 1). 

## 4. Discussion

### 4.1. Autoimmune Thyroid Diseases and Chromosomopathies

#### 4.1.1. Turner Syndrome

Turner syndrome (TS) is caused by a partial or total monosomy of one X chromosome, with or without cell line mosaicism. It is one of the most common chromosomal abnormalities, affecting one in 2000–2500 live-born females. The TS phenotype includes short stature, gonadal dysgenesis, peculiar facial and skeletal dysmorphisms, lymphedema and renal and cardiovascular anomalies [123].

TS girls are at higher risk of developing autoimmune diseases, and AITDs are the most frequent [124]. Other associated autoimmune diseases include: celiac disease (CD), type 1 diabetes mellitus (DMT1), inflammatory bowel diseases (Crohn disease, ulcerative colitis), alopecia areata (AA), vitiligo (V), psoriasis (P), lichen sclerosus (LS) and Addison’s disease (AD) (Table 1). The prevalence of autoimmunity increases with age, and more than one autoimmune disease can coexist in the same patient [33,52,57,100]. Mortensen et al. analyzed several autoantibodies in a large cohort of Danish TS girls and found that anti-TPO and CD autoantibodies coexisted in 9% of cases [41]. Grossi et al. described the association between multiple autoimmune diseases (CD, AA, DMT1, AD) in a TS girl with mosaicism for X monosomy, partial deletion of chromosome 2q and duplication of chromosome 10p [35]. AITDs can also coexist with other endocrinological diseases, such as growth hormone deficiency [50,56,62] and pseudohypoparathyroidism [51].

Several mechanisms have been proposed to explain the increased susceptibility to autoimmune diseases in TS, including haploinsufficiency of X chromosome-related genes. On the X chromosome, there are at least ten genes that are involved in immunoregulatory functions. In particular, haploinsufficiency of the FOXP3 gene, located in the Xp11.23 region, may contribute to increasing the susceptibility of TS girls to AITDs [125]. The higher prevalence of AITDs in women compared to men has led to the consideration of estrogen as a risk factor for these diseases, however, TS girls have lower estrogen exposure than non-TS girls. Bakalov et al. reported an increased prevalence of HT in women with karyotypically normal primary ovarian insufficiency and in women with TS compared with the general male and female populations. The authors suggest that ovarian insufficiency could be a risk factor for AITDs in TS girls and that the haploinsufficiency of X chromosome-related genes may further contribute to increasing the risk of autoimmunity [126]. Some studies have assessed the role of specific genes in the development of autoimmunity in TS, in particular PTPN22 that encodes a lymphoid-specific phosphatase (LYP), which is an important downregulator of T cell activation. PTPN22 C1858T polymorphism has been associated with the development of autoimmunity in a cohort of Brazilian TS subjects [127], but these findings did not emerge in the study performed by Villanueva-Ortega et al. in Mexican TS girls [34].

Additionally, alterations of both humoral and cellular immunity profiles and a lack of Treg immunosuppression have been suggested as possible mechanisms leading to autoimmunity in TS girls. Gawlik et al. evaluated the immunological profile of 37 TS subjects compared with non-TS controls and found lower levels of IgG, lower CD4+ lymphocytes and a significantly lower CD4:CD8 ratio in the TS group [59]; the same results are reported in other studies [128,129,130]. An increased production of proinflammatory cytokines and a decreased production of anti-inflammatory cytokines was found in TS girls by Bakalov et al. [126], but the study of Gawlik et al. did not confirm this alteration [59]. Another interesting result, which emerges from the analysis of Gawlik et al., is the evidence of a lower Treg percentage in TS women with autoimmunity compared with non-TS and TS women without autoimmune disease. However, the results concerning Tregs in the pathogenesis of autoimmunity are inconclusive to date [131,132].

The association between a specific TS karyotype and autoimmune disease is still controversial. According to some authors, patients with an X isochromosome are more prone to develop autoimmunity, especially AITDs [46,59,63,100,122] (Table 1). The study of Mortensen et al. showed that isochromosomal karyotypes have an increased positivity for anti-GAD-65, but not for other autoantibodies [41]. Stoklasova et al. analyzed the relation between karyotypes and autoimmune diseases in a cohort of 286 Czech TS females and found that isolated Xp deletion and triple X mosaicism are related to an increased risk of AITDs [33]. Nevertheless, other authors did not find any evidence of correlation between TS karyotypes and autoimmune diseases [37,39,42,43,44,45,47,48,54,55,57,78,89] (Table 1).

The prevalence rate of AITDs in TS varies widely across studies, probably due to the great difference in age, number and geographical origin of the selected cohorts. However, the prevalence of AITDs in TS girls is higher than in the general population and increases with age [33,37]. The study by Chen et al. [78] showed that thyroid autoantibodies are detected mainly after the age of 10 years, but AITDs can appear even under the age of 2 years [45]. Therefore, thyroid function should be carefully assessed at the time of diagnosis and closely monitored annually, regardless of age [123].

In TS girls, HT is the most common autoimmune disease. The study by El-Mansoury et al. reveals an annual incidence of HT in TS girls of about 3.2% [44]. Aversa et al. evaluated the course of HT in a cohort of 90 TS children and adolescents, compared with non-TS girls and found a lower frequency of family history for HT antecedents, suggesting an inherent predisposition to develop HT in these girls. Further interesting findings were the higher prevalence of euthyroidism and lower median serum levels of TSH and anti-TPO at presentation of the disease [67]. Despite a less severe presentation pattern at diagnosis and irrespective of age or karyotype, TS girls showed a progressive spontaneous deterioration of thyroid function over time, especially in the case of SH at diagnosis (Figure 2). These finding have been confirmed by Wasniewska et al. in two studies which prospectively assessed thyroid function after HT diagnosis in initially euthyroid or SH girls with and without TS. One study evaluated the course of HT in TS and non-TS patients who were initially euthyroid and showed that TS girls exhibited a lower prevalence of both euthyroidism and SH, whereas the prevalence rates of both overt hypothyroidism (47.6%) and hyperthyroidism (4.6%) were significantly higher [54]. A second study evaluated thyroid function, after a 5-year follow-up, in TS girls with HT with an initial biochemical pattern of SH. In more than half of TS girls, the initial thyroid function pattern worsened from SH to overt hypothyroidism (66.7% of cases) (Figure 3) [53].

In addition to changes in function, thyroid structural alterations may also be associated with HT [40]. Calcaterra et al. found that the prevalence of nodular thyroid disease is similar in TS and in non-TS subjects [38]. 

Several reports described the association between GD and TS [49,58,60,61]. As for HT, the prevalence rate of GD in TS girls is higher than in the general population (Table 2). Valenzise et al. showed that the biochemical picture at diagnosis and the clinical course of GD in TS subjects are not different from general pediatric population. In this study, GD prevalence was 1.7%, the methimazole dose required to maintain euthyroidism during the first cycle of therapy, and remission and relapse rates did not significantly differ from those observed in non-TS girls [89]. Finally, GD in TS patients may often be preceded by HT (about 25.7% of cases), regardless of thyroid function and autoimmunity tests at HT diagnosis, as shown by Aversa et al. [111].

To summarise, AITDs in TS, presenting with high prevalence rates, may not be associated with a positive family history of AITDs. With regard to HT, it may have a mild biochemical pattern at presentation, but a progressive deterioration of thyroid function over time and frequent conversion to GD.

#### 4.1.2. Down Syndrome

DS is the most common autosomal aneuploidy, affecting about 1 in 1000 live-born babies [133]. In 95% of cases, the syndrome is due to chromosome 21 trisomy (non-disjunction), while the remaining 5% are either due to Robertsonian translocation or mosaicism [134,135]. 

DS represents the most important genetic cause of intellectual disability, which can be of varying grades, but is usually progressive over time [136]. 

DS patients typically exhibit facial dysmorphisms, hypotonia, congenital heart defects, gastrointestinal abnormalities, increased risk of hematologic malignancies and immunological disorders. Endocrine disorders such as thyroid dysfunction, low bone mass, diabetes, short stature, infertility and propensity to be overweight/obese are much more prevalent than in the general population [84,137]. 

Above all, thyroid dysfunction is the commonest endocrinopathy in DS, ranging from hypothyroidism to hyperthyroidism, either overt or subclinical, of autoimmune or non-autoimmune etiology [95,138]. In particular, congenital hypothyroidism is very common, occurring in about 2.9% of cases [66,85]. With respect to autoimmune diseases, it is well known that DS patients exhibit increased prevalence rates, if compared to the general population, typically of AITDs, AA, V, DMT1, CD and juvenile idiopathic arthritis, which could be in turn associated and sometimes coexist in the same patient [64,68,76,79,80,81,87,88,90,92,94,98,101] (Table 3). 

Several mechanisms have been proposed to explain the increased susceptibility to autoimmunity in DS, which may include: (1) partial central tolerance failure due to altered thymic expression of the autoimmune regulator (AIRE) gene, which is located on chromosome 21and involved in immune regulation [73,83]; (2) thymic atrophy and T and B lymphocyte reduction [74]; (3) increased pro-inflammatory cytokine levels and decreased anti-inflammatory cytokine levels due to alterations in the extracellular adenine nucleotide and nucleoside levels [139,140]. 

AITDs are the most frequently observed autoimmune disorders in DS. The prevalence rates for HT and GD in DS children and adolescents DS are detailed in Table 2.

If compared to the general pediatric population, HT in DS occurs earlier (mean age at diagnosis: 6.5 years), shows no gender predominance, lower frequency of family history for thyroid diseases and higher prevalence of extra-thyroidal autoimmune disorders [72,86,93]. 

HT biochemical presentation in DS seems to be more severe, with a lower frequency of euthyroidism and increased prevalence of SH, which is the most common pattern at diagnosis, followed by overt hypothyroidism [53,65] (Figure 2).

Additionally, the natural history of HT in DS is also unusual, displaying a more severe clinical course and higher rate of conversion to GD over time (about 25% of cases) [82,111] (Figure 3). 

Indeed, DS patients are exposed to a higher risk of thyroid function deterioration over time, which seems to be related to higher baseline TSH levels at diagnosis and to autoimmunity [65]. Nevertheless, it is worth noting that, apart from overt hypothyroidism, much of hypothyroidism in DS appears unrelated to autoimmunity [69]. Thyroid dysfunction—especially the SH pattern, which is commonly observed in very young DS infants—may also be of non-autoimmune etiology, suggesting a congenital thyroid alteration, which is directly related to the trisomy condition of chromosome 21. This phenomenon could be explained by a non-pathological shift in the normal range of TSH as a characteristic of DS, which results in a generally mild and transient form of SH at the group level [65,91,141].

In DS, both HT and GD occur at a younger age than in the general population (between late childhood and early adolescence), without gender preference and frequently in association with other autoimmune disorders [75,77,96,97].

The presenting features of GD at diagnosis seem not to differ in children with or without DS. Instead, the GD clinical course appears to be less severe in DS children and adolescents. According to some authors, GD in DS children shows lower relapse rates during the first methimazole therapy cycle, lower methimazole dosages required to maintain biochemical euthyroidism, higher remission rates after methimazole withdrawal and less need for non-pharmacological treatments [26,77,99]. 

However, this is not a consistent finding, since other authors observed a shorter duration of remission and higher relapse rates with medical treatment in DS patients, suggesting that radioactive iodine treatment may be the best option [97,142]. 

Finally, DS patients are more prone to fluctuations of thyroid function, between hypo- and hyperthyroidism, within the spectrum of thyroid autoimmunity [71]. As mentioned above, the risk of conversion from HT to GD is by far higher in DS children (25.6% vs. 3–4% in non-DS children). In this light, it has been suggested that chromosomopathies might favor the progression from HT to GD. The pathophysiological bases of this predisposition are still unclear, nevertheless, it could be hypothesized that this event reflects the exacerbation of autoimmunity which is typical of patients with DS [26,111]. 

Given the high frequency of thyroid alterations, periodic screening and monitoring of thyroid function and autoimmunity over time are strongly recommended in DS children and adolescents.

#### 4.1.3. Klinefelter Syndrome

Klinefelter syndrome (KS) is the most frequent sex chromosome disorder in men, with the 47, XXY karyotype occurring in about 150 in 100,000 men [102].

Various thyroid abnormalities, including a low response of serum TSH to a thyrotropin-releasing hormone (TRH) stimulation test [143,144], impaired production of thyroxine (T4) by the thyroid gland [102] and lower free triiodothyronine (fT3) serum levels [145], mainly in KS adults, but also in KS children, were described.

Moreover, there were some reports suggesting that males with KS might be at increased risk of certain autoimmune diseases [102,146]. The prevalence of HT was established between 5.4–10%, while GD is very rare [140] (Table 2). Seminog et al., for the first time, revealed that the extra X chromosome could predispose KS patients to a higher prevalence of many autoimmune diseases, particularly those that are predominant in females. Of 30 autoimmune diseases studied in KS subjects, an increased risk for seven autoimmune diseases was confirmed: AD, DMT1, multiple sclerosis, HT, rheumatoid arthritis, Sjogren’s syndrome and systemic lupus erythematosus [147]. Moreover, Panimolle et al. investigated the endocrine organ-specific humoral autoimmunity related to four different organ-specific autoimmune diseases in a group of 47, XXY KS adults and children (DMT1, AD, HT, autoimmune chronic atrophic gastritis). Altogether, humoral organ-specific immunoreactivity was found in 13% of KS patients and its risk progressively increased with the severity of X chromosome polysomy. Furthermore, the frequency of the overall immunoreactivity was higher in KS children than in KS adults. Positivity for at least one of the antibodies investigated was found in 14.5% of KS children, a frequency three times higher than in age-matched controls. The authors suggested the execution of screening for diabetes-, thyroid- and gastric-specific autoimmunity in clinical practice of KS patients [103]. The results of that study might confirm the importance of X-linked gene dosage as a contributing factor for autoimmunity and the influence of sex hormones as modulators of the immune response in KS subjects.

In conclusion, the following points regarding AITDs in KS should be highlighted:-Endocrine organ-specific autoantibodies can be detected in 13% of KS subjects; the frequency progressively increases in those with higher-grade aneuploidies and is higher in children than in adults.-The influence of sex hormones as modulators of the immune response and X-linked gene dosage has been implicated in the pathogenesis of autoimmunity in KS.-In KS patients, measurement of TSH and free T4 (fT4) levels at diagnosis and annually thereafter is suggested.-Screening for thyroid autoantibodies should be considered in KS cases with TSH elevation and/or the presence of goiter or periodically in the absence of suggestive clinical or biochemical signs.

### 4.2. Autoimmune Thyroid Diseases and Deletion Syndromes

#### 4.2.1. Williams Syndrome

Williams syndrome (WS) is a multisystem disorder caused by a microdeletion of chromosome 7q11.23. The deletion is 1.5 to 1.8 Mb and contains 26–28 genes, including the ELN gene that code for “elastin”. WS is a rare disorder, with an estimated incidence of 1:7500 live births and affects boys and girls equally. Common clinical features of WS are facial dysmorphisms (elfin face), cognitive impairment with a friendly and social personality (cocktail party personality), cardiovascular disease (most commonly supravalvular aortic stenosis), renal and genitourinary tract defects [148]. 

Morphological and functional anomalies of the thyroid in WS have been reported by several authors, but association with AITDs is rare. Thyroid hypoplasia is common in WS (about 70% of cases), with the left lobe prevalently involved, and the prevalence increases with age. Additionally, thyroid ectopy and hemiagenesis are described. The mechanism underlying this altered development of the thyroid gland is unclear, and it has been hypothesised that the thyroid gland growth impairment may be linked to a defect of some gene mapped in the 7q11.23 region [149,150]. Recently, BAZ1B, which is deleted in WS, has been linked with the thyroid developmental defects observed in some subjects with WS. BAZ1B deletion is associated with PTEN overexpression and induces an increase in apoptotic phenomena in thyroid cells [151].

With regard to thyroid function, SH is a frequent finding in young children with WS (about 30% of cases), while the prevalence rate of hypothyroidism is about 5–10%. Thyroid function impairment is commonly associated with thyroid hypoplasia, suggesting that the reduction of thyroid volume may be the cause [149,152]. Several authors describe an inverse correlation between SH and age in WS. The youngest WS children shows the highest incidence of SH, with a trend towards normalization in thyroid function with age. This observation may be related to a relative immaturity of the hypothalamic–pituitary–thyroid axis, that can eventually resolve with age [149,153,154]. Hyperthyroidism has also been described [155]. 

AITDs are rarely associated with WS [155] (Table 2). Stagi et al. longitudinally evaluated the prevalence of autoimmune disease in 48 WS subjects. None of the WS patients showed thyroid antibodies [104]. The findings of Stagi et al., in agreement with other reports, suggested that examination of autoimmune diseases should not be performed routinely in asymptomatic WS [149,150,153]. 

Although autoimmune manifestations are infrequent, given the possible association between thyroid disorders and WS, a comprehensive evaluation of thyroid morphology and function should be performed in these patients, especially in younger children.

#### 4.2.2. 22q11.2 Deletion Syndrome

Deletion of chromosome 22q11.2 syndrome (22q11DS) is the most common microdeletion syndrome, with an incidence of about one out of 4000 live births. Phenotypic expression is highly variable and more than 100 different phenotypes have been described, including DiGeorge syndrome (velo-cardio-facial syndrome) [156]. The main clinical features of 22q11DS are congenital cardiac defects, hypoparathyroidism, hypocalcemia, immunodeficiency from thymic hypoplasia, palate anomalies and velopharyngeal dysfunction, facial dysmorphisms and cognitive impairment [157]. 

In 22q11DS, autoimmune diseases occur in about 8% of cases. The most common are juvenile idiopathic arthritis, autoimmune cytopenias, autoimmune skin disorders and AITDs [158]. Additionally, Davies et al. also found a high prevalence of autoimmune complications in 22q11DS athymic patients after thymus transplantation [109]. The mechanism underlying the susceptibility to autoimmune disease is still not completely understood. Impaired immune function, due to abnormal thymic development, may allow self-reactive T cells to survive and initiate autoimmune processes; decreased Tregs may also contribute to the development of autoimmunity [159,160].

Anomalies in thyroid function and structure are frequently associated with 22q11DS. The most common thyroid structural anomalies include: hypoplasia, absence of one lobe, absence of the isthmus and a retroesophageal or retrocarotid extension [160,161,162]. Some authors showed a correlation between thyroid structure alterations, heart malformations and hypoparathyroidism. Genetic studies in mice and mutational analyses in patients have identified T-box 1 (TBX1), mapped on the long arm of chromosome 22 at position 11.21, as a key gene in the pathogenesis of 22q11DS. TBX1 is implicated in cardiac and pharyngeal arch formation and its deletion could be responsible for the cardiac, thymic, thyroid and parathyroid malformations seen in the syndrome [162,163,164]. 

Children with 22q11DS show a high prevalence of both hypo- and hyperthyroidism (about 7.7% and 1.8%, respectively) [163]. 

Choi et al. evaluated the prevalence of endocrine disfunctions in 61 patients with 22q11DS: the most common disease was hypocalcemia (32.8%), followed by overt hypoparathyroidism (13.1%) and short stature (16.4%); AITDs were recorded in 3.28% of patients (1.64% for HT and 1.64% for GD) [105] (Table 2). 

Hypothyroidism seems to be related mainly to thyroid morphology anomalies, and more rarely to HT [162]. 

Several studies described an association between 22q11DS and GD [105,106,108,110]. If compared with the general pediatric population, GD in 22q11DS seems to be more frequent and may occur earlier, even under the age of 3 [107]. According to some authors, certain HLA haplotypes (such as HLA-DR14) and the decreased Treg number may be involved in the early onset of GD in 22q11DS children [106]. 

Kawame et al. reported a case of a 27-month-old 22q11DS female with tachycardia, irritability and seizures due to GD, suggesting that, in addition to the early onset, AITDs in 22q11DS may have an atypical presentation [112]. Additionally, it should be considered that thyrotoxicosis, due to GD, may have a hypercalcemic effect and mask an underling hypoparathyroidism, producing a serum calcium concentration within the normal limits [110]. 

Considering these findings, evaluation of thyroid function, structure and autoimmunity may be considered for 22q11DS children and adolescents [107,108,163].

#### 4.2.3. 18q Deletion Syndrome

Deletions of chromosome 18q syndrome (18qDS) occurs in approximately 1:40,000 live births. The phenotype of the disorder varies greatly between individuals and is not correlated to the size of the deletion. The main clinical features are mental retardation, hypotonia, short stature, flat midface, ear anomalies, abnormal genitalia and foot deformities [165]. 

Autoimmune diseases are described in association with 18qDS, including AITDs, DMT1, juvenile rheumatoid arthritis and pernicious anemia [113,166,167]. Possible explanations of the occurrence of autoimmune diseases in these subjects are the association between 18qDS and immunodeficits, such as IgA and Treg deficiency [113].

Both HT and GD are reported in 18qDS subjects and can occur at an early age. In fact, Tutunculer et al. described GD in a 1-year-old girl with the syndrome [114,115,116]. 

AITDs appear to be common in 18qDS, suggesting that thyroid function screening may be useful in the management of children and adolescents with the syndrome, but further studies are needed to assess the exact incidence of these disorders in this population [114,168].

### 4.3. Autoimmune Thyroid Diseases and Imprinting Disorders

#### Prader–Willi Syndrome

Prader–Willi syndrome (PWS) is a rare genetic disorder resulting from the loss of gene expression within the paternal chromosome 15q11-q13. PWS has a prevalence rate of 1/10,000–30,000 and is characterized by endocrine abnormalities due to hypothalamic–pituitary insufficiency and complex physical, behavioral and intellectual difficulties. Individuals with PWS can present several different endocrine disorders, most of them caused by hypothalamic–pituitary insufficiency [140,169]. The phenotype is likely due to hypothalamic dysfunction, which is responsible for hyperphagia, temperature instability, high pain threshold, hypersomnia and multiple endocrine abnormalities, including growth hormone and TSH deficiencies, hypogonadism and central adrenal insufficiency. Moreover, obesity and its complications are the major causes of morbidity and mortality in PWS [170]. 

Similar to other endocrinopathies, the etiology of hypothyroidism in PWS is thought to be central in origin. The abnormalities of thyroid function are discussed in the literature and published data are discordant. Hypothyroidism has been reported in approximately 20–30% of children with PWS and AITDs are very rare [140,170] (Table 2). In the paper of Iughetti et al., thyroid function tests were carried out in a large population of 339 patients with PWS, aged from 0.2 to 50 years. Hypothyroidism was found to be a frequent feature in subjects with PWS, with a prevalence of 13.6%. Specifically, congenital hypothyroidism was found in 1.18%, acquired overt hypothyroidism in 1.77%, central hypothyroidism in 6.78% and SH in 3.83%. The authors suggested a regular investigation of thyroid function in all PWS patients both at diagnosis and annually during follow-up [171]. Only one study reported slightly positive thyroid autoantibodies (anti-TPO) in one out of 21 PWS patients [117]. Taking into account the higher frequency of hypothyroidism, above all, during the first 2 years of life, a critical period for both growth and development, a special consideration should be given in infancy and early childhood, in order to ensure adequate thyroid hormone levels and, if indicated, to start appropriate levothyroxine therapy.

### 4.4. Autoimmune Thyroid Diseases and RASopathies

#### Neurofibromatosis Type 1 and Noonan Syndrome

Neurofibromatosis type 1 (NF1) is an autosomal dominant tumor predisposition syndrome, caused by loss of function mutations of an anti-oncogene suppressor gene called neurofibromin 1 [172]. Approximately 1:2500 to 1:3500 individuals worldwide are affected, regardless of ethnicity or race. The classic manifestations of NF1 include café-au-lait macules, skinfold freckling, neurofibromas, brain tumors, iris hamartomas and characteristic bony lesions [173]. 

The number of studies reporting an association between NF1 and autoimmune disorders is small but increasing. AITDs have been scarcely described in the literature [140]. Nanda at al. described the case of a 6-year-old boy diagnosed as having NF1 associated with AA and autoimmune thyroiditis [119]. In the study of Guler et al. comprising 78 patients diagnosed with NF1 and 50 healthy controls, the mean levels of fT3, fT4 and TSH were not statistically different between the NF1 and control groups. Similarly, no statistically significant difference was observed between the two groups for anti-TPO and anti-TG positivity [118].

Noonan syndrome (NS) is a multisystem genetic disorder, occurring in 1:1000 to 1:2500 live births. The causative mutations alter genes encoding proteins with roles in the RAS-MAPK pathway, leading to pathway dysregulation. The main clinical features of this syndrome are distinctive facial features, developmental delay, learning difficulties, short stature, congenital heart disease, renal anomalies, lymphatic malformations and bleeding difficulties [174].

In the study of Quaio et al., comprising 42 patients with RASopathies, the majority of whom had NS, 17% had SH without thyroid antibodies, 7% were euthyroid with positive thyroid autoantibodies and another 7% presented overt autoimmune hypothyroidism. Six patients (14%) fulfilled the clinical criteria for autoimmune diseases, including systemic lupus erythematosus, CD, primary antiphospholipid syndrome, autoimmune hepatitis, vitiligo and AITDs [120] (Table 2). 

Moreover, Lee et al. reported a case of an adolescent with NS, cardiac tamponade and HT with clinical and biochemical hypothyroidism, who underwent pericardial drainage [121]. 

According to some authors, screening for AITDs seems to be unnecessary in patients with NF1 [118]. Until a final conclusion on the real incidence of autoimmunity in these syndromes is drawn, pediatricians should be alerted to the possible, although rare, association between AITDs and NS or NF1 [120].

### 4.5. Other Syndromes

In this section, we report an additional genetic disease, McCune–Albright syndrome (MAS). Despite not being included in the systematic review, it is important to mention this syndrome because it represents a relevant cause of hyperthyroidism in children.

MAS is a rare congenital sporadic disorder (the estimated prevalence ranges between 1/100,000 and 1/1,000,000), caused by a postzygotic somatic activating mutation in the GNAS gene encoding the G-protein α subunit (Gsα). MAS is defined by the triad of polyostotic fibrous dysplasia of bone, café-au-lait skin pigmentation and peripheral precocious puberty. Other multiple endocrinopathies—including hyperthyroidism, growth hormone excess, hypercortisolism and renal phosphate wasting—could be associated with the original triad [175]. 

Thyroid disease is the second most common endocrinopathy in patients with MAS, with a reported prevalence of about 31% and onset ranging from 1 to 20 years of age [176,177]. As with all other manifestations of MAS, the activation of the stimulatory G protein is responsible for thyroid gland hyperplasia, thyroid hormone overproduction and increased conversion from T4 to triiodothyronine (T3). Clinically, this results in hyperthyroidism, goiter and/or thyroid nodules [178]. About 64% of MAS patients with thyroid alterations have nodular goiters, with nodules > 1 cm [176]. The functional abnormalities are characterized by autonomous function, frequently with a shifted T3/T4 ratio, suggesting an increase in intra-thyroidal conversion of the pro-hormone T4 into the active metabolite T3. While MAS-associated thyrotoxicosis is not always symptomatic, it is often linked with increased morbidity. The presence of autoimmune pathogenesis in MAS hyperthyroidism has not been documented to date [179]. Moreover, thyroid cancer had been described in a few cases of MAS with demonstrated GNAS mutation in molecular studies, suggesting that MAS might predispose patients to thyroid carcinomas [180].

In conclusion, we must take into account that thyroid disorder with hyperthyroidism, also at a very young age, although not autoimmune, is the second most common endocrinopathy in MAS. For this reason, strict monitoring of thyroid function is recommended every six months in these patients.

## 5. Conclusions

Our analysis confirms that AITDs show peculiar phenotypic patterns when they occur in association with some genetic disorders, especially chromosomopathies. 

To improve clinical practice and healthcare in children and adolescents with genetic syndromes, pediatricians and pediatric endocrinologists should be aware of the importance of the accurate screening and monitoring of thyroid function and autoimmunity in all conditions with an increased incidence of AITDs, such as TS, DS, KS and 22q11DS. Additionally, a careful assessment of thyroid function is recommended in children and adolescents with PWS, WS and MAS, due to a high frequency of hypo- or hyperthyroidism, although it is not autoimmune based. 

Furthermore, maintaining adequate thyroid hormone levels is important in order to not aggravate the growth and cognitive deficits that are not infrequently present in the syndromes analyzed.

Alterations in various genes have been suggested to be responsible for the increased prevalence, the earlier onset or the atypical evolution over time of AITDs in some syndromes. Nevertheless, the underlying pathophysiological mechanisms have not yet been fully elucidated. New technological developments, such as next-generation sequencing technologies, can help identify the causative variants of autoimmune diseases and their association with genetic syndromes. 

## Figures and Tables

**Figure 1 genes-12-00222-f001:**
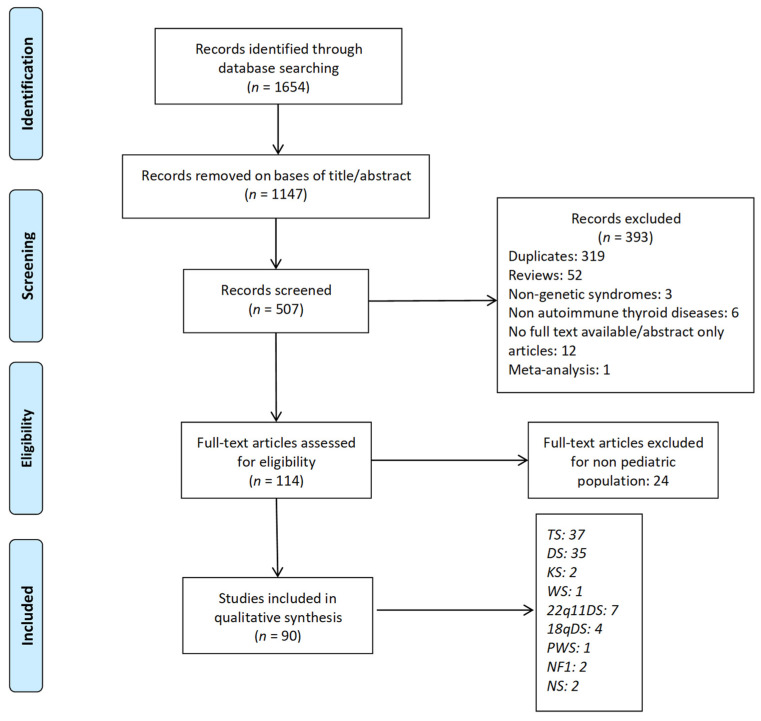
Flow diagram summarizing study selection process. One article [111] is shared by DS and TS. TS: Turner syndrome; DS: Down syndrome; KS: Klinefelter syndrome; 22q11DS: 22q11.2 deletion syndrome; PWS: Prader-Willi syndrome; WS: Williams syndrome; NF1: neurofibromatosis type 1; NS: Noonan syndrome.

**Figure 2 genes-12-00222-f002:**
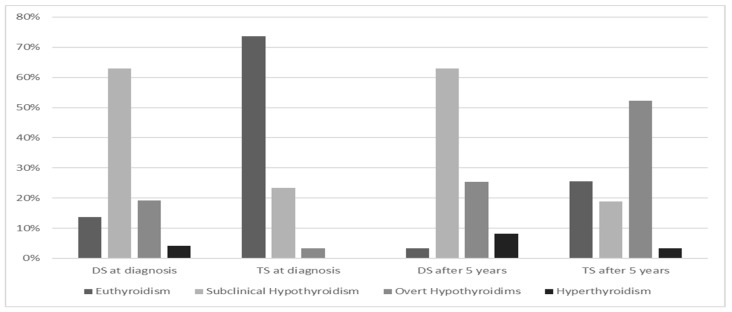
Hashimoto’s thyroiditis presentation pattern and evolution after 5-year follow-up in Turner syndrome (TS) and Down syndrome (DS), according to references [67,72].

**Figure 3 genes-12-00222-f003:**
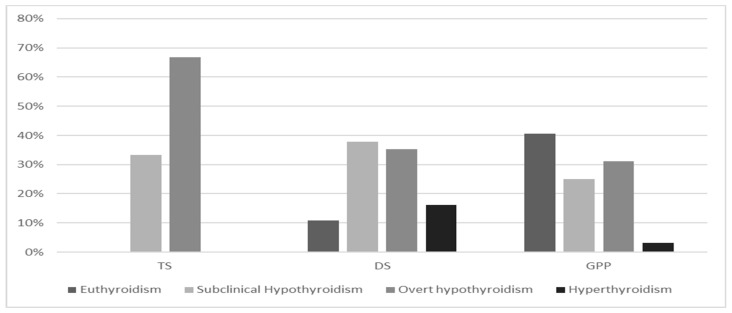
Prevalence (%) of different Hashimoto’s thyroiditis biochemical patterns after 5-year follow-up in Turner syndrome (TS), Down syndrome (DS) and general pediatric population (GPP) with an initial subclinical hypothyroidism presentation, according to references [30,53,65].

**Table 1 genes-12-00222-t001:** Autoimmune thyroid disease (AITD) prevalence, association with karyotype and with other autoimmune diseases, among different studies, in a Turner syndrome population.

Authors, Years	TS Population (n)	TS Population Age Range or Mean (y)	HT Prevalence (%)	GD Prevalence (%)	Autoimmune Associated Disease and Prevalence (%)	Association with AITDs and Specific Karyotype
Stoklasova et al., 2019 [33]	286	2.8–43.3	NA	NA	8.7% CD, 2.1% V, 1.4% AA, 1.4% P, 0.7% IBD	Yes (Xp deletion/triple X mosaicism)
Witkowska-Sędek et al., 2017 [45]	41	6–18	19.5%	NA	NA	No
Chen et al., 2015 [78]	69	0.2–18	10%	4.3%	DMT1	No
Valenzise et al., 2014 [89]	408	1–40	NA	1.7%	DMT1, CD	No
Grossi et al., 2013 [100]	66	1–29.8	21%	0	3% CD, 1.5% AA, 1.5% DMT1, 1.5% V,	Yes (isoXq)
Hamza et al., 2013 [122]	80	4.7–22.3	6.3%	1.3%	12.5% CD, 3.75% anti-GAD-65, 1.25% ACA	Yes (IsoXq)
Kucharska et al., 2013 [36]	54	1–18	20%	0	NA	NA
Gawlik et al., 2011 [37]	86	0–17.4	17%	0	NA	No
Fukuda et al., 2009 [39]	65	15–61	31%	4.6%	NA	No
Medeiros et al., 2009 [40]	17	5.9–22.6	23.5%	5.8%	NA	NA
Mortensen et al., 2009 [41]	107	6–60	15%	1.8%	18% CD, 4% anti-GAD-65	Only for anti-GAD-65 and isoXq
Bettendorf et al., 2006 [42]	120	>16	26%	0	CD	No
Livadas et al., 2005 [43]	84	0.5–19	21%	2.5%	NA	No
El-Mansoury et al., 2005 [44]	91	16–71	25%	2%	NA	No
Elsheikh et al., 2001 [46]	145	16–52	15%	0.7%	NA	Yes (isoXq)
Medeiros et al., 2000 [48]	71	0–19.9	9.8%	NA	NA	No
Hanew et al., 2018 [55]	492	17-42	25.2%	1.8%	1.8% IBD	No
Gawlik et al., 2018 [59]	37	6.3–19.9	NA	NA	CD, V	Yes (isoXq)
Yeşilkaya et al., 2015 [63]	842	0–18	11.1%	0.4%	NA	Yes (isoXq)
Wegiel et al., 2019 [57]	134	0.4–17	14.9%	0	1.5% P, 2.2% V, 0.7% AA, 0.7% LS, 2.7% CD, 1.5% DMT1	No

TS: Turner syndrome, HT: Hashimoto’s thyroiditis, GD: Graves’ disease, CD: celiac disease, V: vitiligo, P: psoriasis, DMT1: diabetes mellitus type 1, AA: alopecia areata, LS: lichen sclerosus, IBD: inflammatory bowel disease, anti-GAD-65: anti-glutamic-acid decarboxylase antibodies, anti-TPO: anti-thyroid peroxidase antibodies, anti-TG: anti-thyroglobulin antibodies, ACA: adrenal cortex autoantibodies, SH: subclinical hypothyroidism, NA: not available.

**Table 2 genes-12-00222-t002:** Prevalence rate of autoimmune thyroid diseases (AITDs) among different genetic syndromes according to the reviewed studies.

Syndrome	HT Prevalence (%)	GD Prevalence (%)	Studies Considered (*n*)
Turner Syndrome	6.3–31%	0.4–5.8%	18 [*]
Down Syndrome	1.4–52.6%	0.7–5.2%	9 [**]
Prader–Willi Syndrome	sporadic	ND	1 [117]
22q11.2 Deletion Syndrome	1.6%	1.6%	1 [105]
Williams Syndrome	ND	ND	1 [104]
RASopathies	7%	ND	1 [120]
Klinefelter Syndrome	7%	ND	1 [102]
Neurofibromatosis Type 1	2.5%	ND	1 [118]

HT: Hashimoto’s thyroiditis, GD: Graves’ disease, ND: no data. * according to the references [36,37,39,40,41,42,43,44,45,46,48,55,57,63,78,89,100,122] ** according to the references [65,68,69,73,74,82,83,85,88].

**Table 3 genes-12-00222-t003:** Autoimmune thyroid disease (AITD) prevalence and association with other autoimmune diseases, among different studies, in Down syndrome population.

Author, Year	DS Population (*n*)	DS Population Age Range or Mean (y)	HT Prevalence (%)	GD Prevalence (%)	Autoimmune-Associated Diseases and Prevalence (%)
Pepe et al., 2020 [65]	101	2–17	36.6%	NA	4% CD, 2% DMT1, 1% V
Zwaveling-Soonawala et al., 2017 [88]	123	10.7 (mean)	21.9%	0.7%	4% CD
Aversa et al., 2016 [70]	174	1–18	NA	NA	14.3% CD, 4% DMT1, 27% AA, 13% V
Giménez-Barcons et al., 2014 [73]	19	0–10	52.6%	5.2%	5.2% CD
Pellegrini et al., 2012 [74]	29	1.4–22.8	17.2%	3.4%	3.5% V, 10.5% CD
Unachak et al., 2008 [82]	140	0-14	1.4%	1.4%	NA
Soderbergh et al., 2006 [83]	48	11–56	18.7%	0	4% CD, 4% AA, 12.5% APS1 related antibodies
Pierce et al., 2017 [69]	508	0.05–26	5.1%	1.6%	3.7% CD, 0.8% DMT1
Gruñeiro de Papendieck et al., 2002 [85]	137	0.04–16	15.3%	2.9%	NA
AlAaraj et al., 2019 [66]	102	2.3 ± 3 (mean)	NA	NA	1.96% DMT1
Abdulrazzaq et al., 2018 [68]	92	<18	14.1%	NA	4.3% DMT1, 1.1% CD

DS: Down syndrome, HT: Hashimoto’s thyroiditis, GD: Graves’ disease, CD: celiac disease, V: vitiligo, DMT1: diabetes mellitus type 1, AA: alopecia areata, anti-TPO: anti-thyroid peroxidase antibodies, anti-TG: anti-thyroglobulin antibodies, APS1: autoimmune polyendocrine syndrome type I, SH: subclinical hypothyroidism, NA: not available.

## Data Availability

Not applicable.

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
