# Peer review of "Hashimoto’s Thyroiditis and Graves’ Disease in Genetic Syndromes in Pediatric Age"

_genes, 2021, doi:10.3390/genes12020222_

Round 1
Reviewer 1 Report
This paper is well described review, enough to publish in present form.
Author Response
We thank the Referee for such high consideration of our manuscript.
Reviewer 2 Report
Very thorough review. I would only suggest that you explain why you only searched for papers on those particular syndromes.
Author Response
Response to Reviewer 2 Comments
We are grateful to the Referee for the very constructive revision of our manuscript.
Point 1: Very thorough review. I would only suggest that you explain why you only searched for papers on those particular syndromes.
Response 1: We chose to focus our review on the genetic syndromes more frequently associated with Autoimmune thyroid diseases (AITDs), based on the available literature data.
As suggested, we explained this point in the Materials and Methods section (page 3, lines 135-138): “From our initial research, all the genetic syndromes most frequently associated with AITDs were subsequently reviewed individually, hence we focused on: Turner Syndrome, Klinefelter syndrome, Down Syndrome, 18q Deletion Syndrome, Prader-Willi Syndrome, 22q11.2 Deletion Syndrome, Neurofibomatosis type 1, Noonan Syndrome, Williams Syndrome”.
Reviewer 3 Report
This is a study examining association of autoimmune thyroid disease in various genetic syndromes. The authors did the systemic review for this particular issues. I have the following comments/suggestions.
- Line 61: Please include the locus of chromosome for TSHR
- Line 83, 84, 89: Please include the locus of chromosome for CD25, CD40, CTLA-4, and PTPN22
- Line 98: 1,2% --is this 1.2 or 1-2 or 12?
- Figure 1: the bottom of diagram, n=90 does not match with the total of genetic syndromes (I added up the number of genetic syndromes and it's 91)
- Line 161: The sentence is confusing. You wrote TS girls, then why you also wrote "both male and female"?
- Table 1 and Table 2: In table 1, many studies include subjects age >18 years. Is the prevalence in Table 2 include adult population?
- Line 297: Please put the full name for AIRE
- Line 324: Line 324, consider revising "As well as HT, also GD....". May be "Both HT and GD"?
- Line 354: please put reference for incidence of 47, XXY
- Line 458-459: 3.3% is not equal to 1.6% (HT) and 1.6% (GD)
- Line 524: Use levothyroxine
- Line 133: Do you mean "which included only non-pediatric"? Otherwise you should exclude all studies that included non-pediatric population.
Author Response
We thank the Referee for such valuable comments.
Please see the attachment.

Round 2
Reviewer 2 Report
Nice work. I have no additional suggestions.
Reviewer 3 Report
The revised version looks good!